# The association of SARS-CoV-2 infection and tuberculosis disease with unfavorable treatment outcomes: A systematic review

Nirma Khatri Vadlamudi[1], C. Andrew Basham[2], James C. Johnston[3], Faiz Ahmad Khan[4,5], Giovanni Battista Migliori[6], Rosella Centis[6], Lia D'Ambrosio[7], Waasila Jassat[8,9], Mary-Ann Davies[10,11], Kevin Schwartzman[4,5,12], Jonathon R. Campbell[4,5,13] *

1 Faculty of Medicine, Department of Pediatrics, The University of British Columbia, Vancouver, Canada, 2 Division of Pharmacoepidemiology and Pharmacoeconomics, Brigham and Women's Hospital and Harvard Medical School, Boston, Massachusetts, United States of America, 3 Faculty of Medicine, Department of Medicine, The University of British Columbia, Vancouver, Canada, 4 Research Institute of the McGill University Health Centre, Respiratory Epidemiology and Clinical Research Unit, Centre for Outcomes Research & Evaluation, Montreal, Canada, 5 McGill International TB Centre, Montreal, Canada, 6 Istituti Clinici Scientifici Maugeri IRCCS, Servizio di Epidemiologia Clinica delle Malattie Respiratorie, Tradate, Italy, 7 Public Health Consulting Group, Lugano, Switzerland, 8 National Institute for Communicable Diseases (NICD) of the National Health Laboratory Service, Division of Public Health Surveillance and Response, Johannesburg, South Africa, 9 Right to Care, Pretoria, South Africa, 10 Western Cape Government, Health and Wellness, Cape Town, South Africa, 11 School of Public Health and Family Medicine, University of Cape Town, Cape Town, South Africa, 12 Respiratory Division, McGill University, Montreal, Canada, 13 Department of Medicine & Department of Global and Public Health, McGill University, Montreal, Canada

* jonathon.campbell@mcgill.ca

## Abstract

### Background

Whether SARS-CoV-2 infection and its management influence tuberculosis (TB) treatment outcomes is uncertain. We synthesized evidence on the association of SARS-CoV-2 coinfection (*Coinfection Review*) and its management (*Clinical Management Review*) on treatment outcomes among people with tuberculosis (TB) disease.

### Methods

We systematically searched the literature from 1 January 2020 to 6 February 2022. Primary outcomes included: unfavorable (death, treatment failure, loss-to-follow-up) TB treatment outcomes (*Coinfection* and *Clinical Management Review*) and/or severe or critical COVID-19 or death (*Clinical Management Review*). Study quality was assessed with an adapted Newcastle Ottawa Scale. Data were heterogeneous and a narrative review was performed. An updated search was performed on April 3, 2023.

### Findings

From 9,529 records, we included 11 studies and 7305 unique participants. No study reported data relevant to our review in their primary publication and data had to be contributed by study authors after contact. Evidence from all studies was low quality. Eight studies of 5749 persons treated for TB (286 [5%] with SARS-CoV-2) were included in the

**Funding:** This work was supported by the World Health Organization (2022/1207232-0 to JRC). The funders had no role in study design, data collection and analysis, decision to publish, or preparation of the manuscript.

**Competing interests:** The authors have read the journal's policy and have the following competing interests: NKV reports receiving consulting fees from Broadstreet HEOR for unrelated projects outside of the submitted work. JRC reports receiving consulting fees from the COVID-19 Immunity Task Force (Canada) and The World Bank, for unrelated projects outside of the submitted work. This does not alter our adherence to PLOS policies on sharing data and materials.

*Coinfection Review.* Across five studies reporting our primary outcome, there was no significant association between SARS-CoV-2 coinfection and unfavorable TB treatment outcomes. Four studies of 1572 TB patients—of whom 291 (19%) received corticosteroids or other immunomodulating treatment—were included in the *Clinical Management Review*, and two addressed a primary outcome. Studies were likely confounded by indication and discordant findings existed among studies. When updating our search, we still did not identify any study reporting data relevant to this review in their primary publication.

## Interpretation

No study was designed to answer our research questions of interest. It remains unclear whether TB/SARS-CoV-2 and its therapeutic management are associated with unfavorable outcomes. Research is needed to improve our understanding of risk and optimal management of persons with TB and SARS-CoV-2 infection.

## Trial registration

**Registration**: PROSPERO (CRD42022309818).

## Background

Tuberculosis (TB) was the leading cause of death globally due to a single infectious agent prior to the emergence of the severe acute respiratory syndrome coronavirus 2 (SARS-CoV-2) pandemic [1]. As of April 14, 2023, there were over 6.9 million recorded deaths attributable to COVID-19, the disease caused by SARS-CoV-2. In parallel, 1.6 million people die each year from TB [2,3].

The SARS-CoV-2 pandemic has stressed health systems in many countries—often at the expense of continuity and quality of TB services [4]. Indeed, evidence from multiple countries suggests substantial reductions in TB diagnosis and, after over a decade of consistent declines, the WHO estimated that TB deaths increased in 2020 and 2021 [4,5].

Beyond the impact of the pandemic on TB programming, it is unknown whether people co-infected with TB and SARS-CoV-2 may have worse prognoses than those with only TB [4,6]. Furthermore, there is limited information about the therapeutic value of corticosteroids or other immunomodulating treatments (e.g., tocilizumab, baricitinib) among people with SARS-CoV-2 and TB coinfection. These data would provide critical evidence to inform decisions around testing persons with TB for SARS-CoV-2, prioritizing persons with TB for SARS-CoV-2 vaccination, and administration of SARS-CoV-2 therapeutics among people with SARS-CoV-2 and TB coinfection.

Given these uncertainties, we conducted a systematic review to inform the WHO revision of clinical guidance for COVID-19 treatment aiming to synthesize evidence on SARS-CoV-2 and TB coinfection to establish (1) if persons treated for TB disease have worse TB-related outcomes when coinfected with SARS-CoV-2 (*Coinfection Review*); and (2) if corticosteroids or other immunomodulating treatments improve TB-related and COVID-19-related outcomes for persons with TB disease and SARS-CoV-2 coinfection (*Clinical Management Review*).

## Methods

### Search strategy and study selection criteria

This systematic review and meta-analysis followed the Preferred Reporting Items for Systematic Reviews and Meta-analyses (PRISMA) guidelines [7] and was prospectively registered on PROSPERO (CRD42022309818). As this study was an aggregate data systematic review, ethical approval was not required.

We searched MEDLINE, Embase, Web of Science, Scopus, medRxiv, bioRxiv, and the WHO library for eligible studies. The searches were executed between February 3–6, 2022, and included citations from Jan 1, 2020, to the date of search (except for medRxiv/bioRxiv, which had a start date of March 1, 2020). All searches were designed in consultation with an experienced medical librarian. We used a combination of free text terms and MeSH terms containing concepts related to SARS-CoV-2, COVID-19, and tuberculosis (see **Table A in S1 Text**). There was no limit on language or geographical locations. Before screening, we narrowed preprint literature to only manuscripts that contained the words "COVID" or "SARS" and "tuberculosis" or "TB" in the title and/or abstract.

Study eligibility was defined *a priori* according to the PECO (population, exposure, comparator, outcome) framework (**Table B in S1 Text**). We included randomized controlled trials, cohort, and case-control studies published in any language with an English abstract available that (1) compared TB treatment outcomes among those who tested positive for SARS-CoV-2 compared with those who tested negative for SARS-CoV-2 and/or (2) compared TB and/or COVID-19 outcomes among those with SARS-CoV-2 and TB coinfection treated with vs. without corticosteroids or other immunomodulating treatments. We excluded cross-sectional studies, opinions, letters to the editor, and modeling analyses, and those without human participants. Two investigators (CAB and NKV) screened study titles and abstracts—any study flagged as relevant by at least one reviewer was included for full text screening. Discordance between reviewers during full text screening was resolved by discussion or by a third reviewer (JRC). For included studies, we manually searched reference lists for other relevant studies.

### Data extraction and quality assessment

We did not identify any studies that provided adequate data for inclusion in our review. However, we noted several with potentially relevant data available not reported in the primary publication; we contacted these authors. Study data contained in primary publications were extracted by two independent investigators (CAB and NKV) in a blinded manner [8], while other data were provided by study authors. Data included study population, location, design, dates of recruitment and follow-up, demographic and clinical characteristics, diagnosis methods, TB and SARS-CoV-2 treatments provided, and study outcomes (as relevant).

Risk of bias was assessed independently by two investigators using an adapted Newcastle-Ottawa Scale (NOS) for cohort and case-control studies (no randomized trials were included) [9]. The NOS tool had nine domains assessing risk of selection bias, degree and extent of comparability between exposed and comparator populations, and risk of bias in ascertainment of exposures and/or outcomes (**Table C in S1 Text**). We considered a study receiving a score of 7 or higher to be "high quality" unless a critical error in comparability was noted, in which case it was downgraded to low quality. Discrepancies in data extraction or quality assessment were resolved through discussion or by a third investigator (JRC).

## Data synthesis and analyses

For the *Coinfection Review* (comparing persons with TB with vs. without SARS-CoV-2 infection), the primary outcome was unfavorable end of TB treatment outcomes (i.e., loss to follow-up, treatment failure, or all cause mortality during treatment). For the *Clinical Management Review* (comparing persons with TB and SARS-CoV-2 infection receiving vs. not receiving corticosteroids or immunomodulating treatments), the co-primary outcomes were (i) unfavorable end of TB treatment outcomes and (ii) a composite outcome of severe or critical COVID-19 or death. Traditional definitions of severe or critical COVID-19 [10] were inconsistently used in studies, so surrogate markers, i.e., admission to the intensive care unit and/or use of mechanical ventilation, were employed to define these outcomes. Secondary outcomes for both reviews included: all-cause mortality, loss to follow-up, treatment failure or recurrence, microbiologic conversion of TB (positive-to-negative) after two months of treatment, hospitalization and its duration, sequelae, and medication adherence. For the *Clinical Management Review*, secondary outcomes also included mechanical ventilation, use of vasopressors, renal replacement therapy, coagulopathies, and symptoms consistent with long COVID.

For each study, we estimated the relative risk for each primary and secondary outcome through odds ratios and 95% confidence intervals (95% CI) calculated with 2-by-2 tables or author-supplied estimates from regression models. We estimated the relative risk of primary and secondary outcomes for the total population included and for three subgroups based on timing of SARS-CoV-2 infection in relation to TB treatment initiation, as this may modulate associations: (i) SARS-CoV-2 infection diagnosed >28 days to 6 months prior to TB treatment initiation; (ii) SARS-CoV-2 infection diagnosed ≤28 days before and ≤28 days after TB treatment initiation; and (iii) SARS-CoV-2 infection diagnosed >28 days after TB treatment initiation but before TB treatment was terminated. In addition, we estimated the relative risk of primary outcomes for additional subgroups defined by sociodemographic and clinical characteristics (e.g., HIV coinfection), TB-related characteristics, SARS-CoV-2-related characteristics, and study-level characteristics (**Table B in S1 Text**).

As data were too heterogeneous, we limited our review to a narrative synthesis following Synthesis Without Meta-Analysis (SWiM) reporting guidelines [11]. To qualify our certainty in the evidence, we used the Grading of Recommendations Assessment, Development and Evaluation (GRADE) framework and generated evidence profiles and summary of findings tables [12].

## Results

### Search strategy and characteristics of included studies

We identified 9,529 records; 5,513 remained after removing duplicates. A further 5,472 records were excluded after screening titles and abstracts, leaving 41 unique full texts. Of these 41 full texts, we contacted 21 authors of studies with potentially relevant data, of whom 16 (76%) replied and ultimately 11 (52%) shared data (one author provided data from a different report; **Fig 1**) [13–23]. Full details on excluded studies are described **Table D in S1 Text**. Overall, seven studies were included in the *Coinfection Review* [13–19], three in the *Clinical Management Review* [20–22], and one in both reviews [23].

The characteristics of the 11 included studies are reported in **Table 1** (original author-supplied data in **S2 Text**). Overall, 7,305 unique participants were included with a median (range) number of participants per study of 139 (20 to 5,409). All included studies were retrospective in design (nine cohort, two case-control, and one multi-study analysis with retrospective designs) [13–23]. Four of these studies were conducted in South Africa [14,21–23], one was a

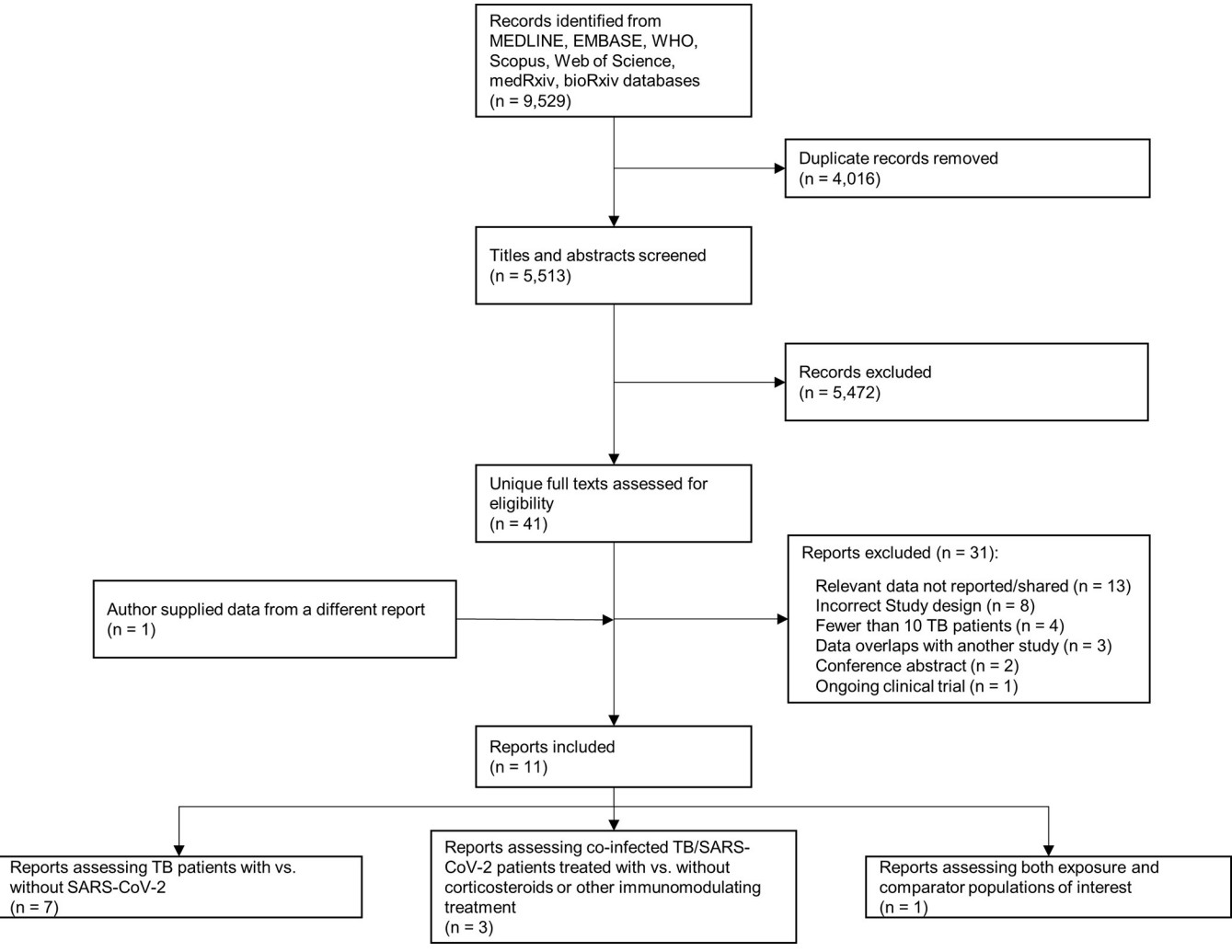

**Fig 1. PRISMA flow diagram.**

global study involving 24 countries [20], while one study each was performed in Turkey [13], Italy [15], Russia [16], Brazil [17], Indonesia [18], and India [19]. Most (n = 8) were conducted among persons in hospital settings [13,15–18,21–23]. When known [14–17,20,23], the majority (72–96%) of TB was pulmonary; TB was predominantly rifampicin-resistant in two studies [14,17]. HIV coinfection was present in >50% of included participants in five of eight studies reporting such data [14,15,17,18,20,21,23]. Most (n = 10) studies were in adults [13–15,17–23], while one was in children (median age of 9 years, range 3 to 12 years) [16]. Two studies contained relevant data on circulating SARS-CoV-2 variants [21,22] and one had data on SARS-CoV-2 vaccination status of participants [22] (5 studies recruited participants exclusively in 2020, while 6 studies recruited participants in 2021/2022, though only 2 studies beyond June 2021).

All included studies were classified as low quality for the purpose of answering our review questions (**Fig 2**). Notably, all studies were at high risk of bias with respect to comparability/ exchangeability of exposure and comparator groups and potential for confounding. Details on management and care offered to persons in each group were often limited, leaving unclear risk of bias due to potential differences in management and care offered to exposure and

**Table 1. Characteristics of included studies.**

| Author (Year) | Study Design | Country | Setting | Study Period | Number of Participants | TB Characteristics | HIV-Coinfection | Sex Distribution | Age of Participants (Years) |
|---|---|---|---|---|---|---|---|---|---|
| **TB Patients Co-infected with SARS-CoV-2 vs. TB Patients without SARS-CoV-2 (*Coinfection Review*)** | | | | | | | | | |
| Kilic (2022) [13] | Retrospective Cohort | Turkey | Outpatient | November 2019 to April 2020 | 20 participants in total; 4 TB/ SARS-CoV-2 Co-infected, 16 only TB | U | U | Co-infected: 25% female Only TB: 12% female | Median Age of: Co-infected = 45 Only TB = 53 |
| Mohr-Holland (2021) [14] | Retrospective Cohort | South Africa | Outpatient and Hospital | March 2020 to June 2021 | 139 participants in total; 36 TB/ SARS-CoV-2 Co-infected, 103 only TB | 123 (88%) pulmonary TB; 139 (100%) RR-TB | 103 (74%) living with HIV | 48% female | Median: 35 (IQR 30–48) |
| Stochino† (2020) [15] | Retrospective Cohort | Italy | Hospital | March to April 2020 | 24 participants in total; 20 TB/ SARS-CoV-2 Co-infected, 4 only TB | 23 (96%) pulmonary TB; 3 (13%) RR-TB | 1 (4%) living with HIV | Co-infected: 40% female Only TB: 50% female | Median Age of: Co-infected = 39 Only TB = 27 |
| du Bruyn (2021) [23] | Retrospective Case-Control | South Africa | Hospital | June to August 2020 | 20 participants in total; 15 TB/ SARS-CoV-2 Co-infected, 5 only TB | 2 (10%) RR-TB | 13 (65%) living with HIV | 52% female | Median Age of: Co-infected = 37 Only TB = 42 |
| Gubkina (2020) [16] | Retrospective Cohort | Russia | Hospital | April to July 2020 | 25 participants in total; 8 TB/ SARS-CoV-2 Co-infected, 17 only TB* | 18 (72%) pulmonary TB | U | 76% female | Median: 9 (Range: 3–12) |
| Gomes (2021) [17] | Retrospective Cohort | Brazil | Hospital | September 2020 to February 2021 | 83 participants in total; 3 TB/ SARS-CoV-2 Co-infected, 80 only TB** | 73 (88%) pulmonary TB; 48 (58%) RR-TB | 6 (7%) living with HIV | 40% female | Median: 55 |
| Zulmansyah (2021) [18] | Retrospective Cohort | Indonesia | Hospital | June to November 2020 | 29 participants in total; 16 TB/ SARS-CoV-2 Co-infected, 13 only TB | U | 0 (0%) living with HIV | "Mostly male" | Range: 18–86 |
| Kumar (2021) [19] | Retrospective Cohort | India | Outpatient and Hospital | October 2020 to March 2021 | 5409 participants in total; 184 TB/ SARS-CoV-2 Co-infected, 5225 only TB | U | U | 34% female | Median: 50 |
| **TB Patients Co-Infected With SARS-CoV-2 Receiving Corticosteroids or Other Immunomodulating Treatments vs. TB Patients Co-Infected with SARS-CoV-2 Not Receiving These Treatments (*Clinical Management Review*)** | | | | | | | | | |

*(Continued)*

**Table 1.** (Continued)

| Author (Year) | Study Design | Country | Setting | Study Period | Number of Participants | TB Characteristics | HIV-Coinfection | Sex Distribution | Age of Participants (Years) |
|---|---|---|---|---|---|---|---|---|---|
| Jassat (2021) [21] | Retrospective Cohort | South Africa | Hospital | October 2020 to April 2022 | 806 participants in total; 117 receiving steroid or IM treatment, 689 no treatment | U | 435 (54%) living with HIV | Treatment = 47% female  No Treatment = 45% female | Median Age of: Treatment = 41  No Treatment = 44 |
| The TB/ COVID-19 Global Study Group (2022) [20] | Multi-Study Analysis (Retrospective Designs) | Multi-country (34 total) | Outpatient and Hospital | 2020–2021 | 118 participants in total;*** 23 receiving steroid or IM treatment, 95 no treatment | 99 (84%) pulmonary TB; 16 (14%) DR-TB | 16 (14%) living with HIV | 30% female | Median: 44 (IQR 31–58) |
| du Bruyn (2021) [23] | Retrospective Case-Control | South Africa | Hospital | June to August 2020 | 15 participants in total; 8 receiving steroids or IM treatment, 7 no treatment | 2 (13%) RR-TB | 8 (53%) living with HIV | U | Median: 37 |
| Davies (2022) [22] | Retrospective Cohort | South Africa | Hospital | November 2020 to January 2021 | 633 participants in total; 143 receiving steroid or IM treatment, 490 no treatment | U | 390 (61.1%) living with HIV | Treatment = 47% female  No Treatment = 51% female | Median Age of: Treatment = 41  No Treatment = 40 |

Abbreviations: RR-TB, rifampicin-resistant tuberculosis; TB, tuberculosis; DR-TB, drug-resistant tuberculosis; HIV, human immunodeficiency virus; SARS-CoV-2, severe acute respiratory syndrome coronavirus 2; COVID-19, coronavirus disease 2019; U, unknown; IM, immunomodulating.

*16/17 (94%) participants with only TB had evidence of previous SARS-CoV-2 infection via serological test

**21/80 (26%) participants with only TB had evidence of previous SARS-CoV-2 infection via serological test

***Of the 118 participants, 5 (4%) had SARS-CoV-2 diagnosed >28 days prior to TB, 52 (44%) had SARS-CoV-2 diagnosed within 28 days of TB, and 61 (52%) had SARS-CoV-2 diagnosed >28 days after initiating TB treatment; †One patient in this cohort is also reported in The TB/COVID-19 Global Study Group cohort, however they are not double counted in analyses.

comparator groups in many studies. Nearly all studies were at high risk of bias due to representativeness of the exposure group. Studies in the *Clinical Management Review* were at high risk of misclassification bias, primarily because documentation of receipt of corticosteroids or other immunomodulating treatment was not systematic. There was also high risk of bias with respect to the presence or absence of the primary outcome at enrollment, as timing of steroid or immunomodulating treatment initiation was unknown and may have occurred after an outcome of interest (e.g., critical COVID-19), and high risk for confounding by indication, as sicker participants were generally more likely to receive corticosteroids or other immunomodulating treatment. Additional detail is provided in S3 Text.

## Coinfection review

**Study characteristics.** A total of 8 studies consisting of 5749 participants were included (range: 20 to 5409 participants) [13–19,23]. Among the included participants, 286 (5%) had TB and SARS-CoV-2 infection, while 5463 (95%) had TB disease alone. Three studies (38%) recruited participants in 2021 [14,17,19]. One study focused on children [16], while the

| Study | Review Question* | Risk of Bias Related to Selection and Group Assignment | | | | Risk of Bias Related to Comparability of Populations | | Risk of Bias Related to Outcome Assessment and/or Ascertainment | | |
|---|---|---|---|---|---|---|---|---|---|---|
| | | Representativeness of Exposure Group | Representativeness of Comparator Group | Misclassification | Primary Outcome not Present at Enrolment | Group Similarity / Confounding | Similarity in Management and Care | Outcome Ascertainment | Follow-up Duration | Loss to Follow-Up Rate |
| Davies (2022) [22] | Clin Manag | High | High | High | Unclear | High | Unclear | Low | High | Low |
| du Bruyn (2021) [23] | Coinfection | High | Low | Low | Low | High | Low | Low | High | Low |
| | Clin Manag | High | Low | High | High | High | Low | Low | Low | Low |
| Gomes (2021) [17] | Coinfection | High | Unclear | Low | Low | High | Low | Low | Low | Low |
| Gubkina (2020) [16] | Coinfection | High | Low | Low | Low | High | Low | Low | Unclear | Low |
| Jassat (2021) [21] | Clin Manag | High | Low | High | Unclear | High | Unclear | Low | High | Low |
| Kilic (2022) [13] | Coinfection | Low | Low | Unclear | Low | High | Unclear | Low | Low | Low |
| Kumar (2021) [19] | Coinfection | High | Low | High | Unclear | High | Unclear | Unclear | Unclear | Low |
| The TB/COVID-19 Global Study Group (2022) [20] | Clin Manag | High | Unclear | Low | Unclear | High | Unclear | Unclear | Low | High |
| Mohr-Holland (2021) [14] | Coinfection | High | Unclear | Low | Low | High | Unclear | Low | Low | High |
| Stochino (2020) [15] | Coinfection | High | Low | Low | Low | High | Unclear | Low | Low | High |
| Zulmansyah (2021) [18] | Coinfection | High | Low | Unclear | Low | High | Unclear | Low | High | Low |

*Coinfection refers to a study evaluating if individuals with TB have worse treatment outcomes if they also have SARS-CoV-2 infection; *Clin Manag* refers to a study evaluating if individuals with TB/SARS-CoV-2 coinfection have worse outcomes if they receive vs. don't receive steroids or immunomodulating treatments.

**Fig 2. Risk of bias assessment.**

remainder involved adults [13–15,17–19,23]. Participants were predominantly male, driven mostly by one study consisting of 5409 participants of whom 66% were male (**Table 1**) [19]. Among the eight studies in this review, corticosteroids or other immunomodulating treatments were not used in five studies, use was unknown in two studies [14,17,19], and were used by 8/15 (53%) of participants with TB and SARS-CoV-2 infection in one study [23].

**Primary outcomes.** The primary outcome of interest (any unfavorable TB treatment outcome) was assessed in five (63%) included studies [13–17], comprising 203 participants, of whom 93 (46%) were coinfected or previously infected with SARS-CoV-2. The proportion of unfavorable TB treatment outcomes among coinfected participants ranged from 0% to 67%, and among participants with only TB disease from 0% to 63% (**Table 2**). Crude estimates of the odds ratio (where an odds ratio <1 suggests SARS-CoV-2 coinfection reduces unfavorable outcomes and an odds ratio >1 suggests SARS-CoV-2 coinfection increases unfavorable outcomes) for unfavorable TB treatment outcomes suggested no significant association with TB and SARS-CoV-2 infection (**Table 2**). Point estimates for odds ratios in the included studies ranged from 0.36 to 1.29. In two studies [13,16], no unfavorable outcomes occurred in either group. Few subgroup analyses were possible with the available data and results remained heterogeneous (see **S3 Text**).

**Secondary outcomes.** Few studies reported the secondary outcomes identified for this review. All-cause mortality was reported in six studies [13–17,19], while in-hospital mortality (for studies where observation of participants ended at hospital discharge or death) was reported in two studies (**Table E in S1 Text**) [18,23]. In most studies, there was no significant difference in all-cause mortality between persons treated for TB with vs. without SARS-CoV-2 coinfection. With respect to in-hospital mortality, deaths only occurred in the group with SARS-CoV-2 coinfection in the two studies reporting this outcome [18,23]. Loss to follow-up and treatment failure were relatively uncommon across studies, with no trends to suggest these differed systematically between groups (**Table F and Table G in S1 Text**) [13–17]. In the one study that reported length of hospitalization [18], it was 17.25 days for coinfected participants, compared to 12.89 days for participants with only TB disease; this difference was statistically significant after adjusting for age, sex, and other co-morbidities (p = 0.048).

## Clinical management review

**Study characteristics.** A total of 4 studies with 1,572 participants were included (range: 15 to 806 participants) [20–23]. Among these participants, 291 (19%) received corticosteroids or

**Table 2. Outcomes for select populations in each study evaluating TB patients co-infected with SARS-CoV-2 vs. TB patients without SARS-CoV-2 (Coinfection Review).**

| Author (Year) | Population Subgroup | Outcome | Participants with Known Outcome | Participants with Outcome (%) | Odds Ratio (95% CI) for Outcome if Co-Infected with SARS-CoV-2 |
|---|---|---|---|---|---|
| Kilic (2022) [13] | All Participants (All SARS-CoV-2 diagnosed >28 days after TB) | All Unfavorable TB Outcomes | TB and SARS-CoV-2 Co-Infected: 4 TB Only: 16 | TB and SARS-CoV-2 Co-Infected: 0 (0%) TB Only: 0 (0%) | N/A |
| Mohr-Holland (2021) [14] | All Participants | All Unfavorable TB Outcomes | TB and SARS-CoV-2 Co-Infected: 27 TB Only: 54 | TB and SARS-CoV-2 Co-Infected: 18 (67%) TB Only: 34 (63%) | 1.18 (0.44 to 3.11) |
| | Only SARS-CoV-2 diagnosed within 28 days of TB | All Unfavorable TB Outcomes | TB and SARS-CoV-2 Co-Infected: 7 TB Only: 54 | TB and SARS-CoV-2 Co-Infected: 4 (57%) TB Only: 34 (63%) | 0.78 (0.16 to 3.87) |
| | Only SARS-CoV-2 diagnosed >28 days after TB | All Unfavorable TB Outcomes | TB and SARS-CoV-2 Co-Infected: 20 TB Only: 54 | TB and SARS-CoV-2 Co-Infected: 14 (70%) TB Only: 34 (64%) | 1.37 (0.45 to 4.14) |
| | People living with HIV | All Unfavorable TB Outcomes | TB and SARS-CoV-2 Co-Infected: 18 TB Only: 41 | TB and SARS-CoV-2 Co-Infected: 13 (72%) TB Only: 26 (63%) | 1.50 (0.45 to 5.04) |
| | People without HIV | All Unfavorable TB Outcomes | TB and SARS-CoV-2 Co-Infected: 9 TB Only: 13 | TB and SARS-CoV-2 Co-Infected: 5 (56%) TB Only: 8 (62%) | 0.78 (0.14 to 4.39) |
| Stochino (2020) [15] | All Participants | All Unfavorable TB Outcomes | TB and SARS-CoV-2 Co-Infected: 20 TB Only: 4 | TB and SARS-CoV-2 Co-Infected: 6 (30%) TB Only: 1 (25%) | 1.29 (0.11 to 15.00) |
| | Only SARS-CoV-2 diagnosed within 28 days of TB | All Unfavorable TB Outcomes | TB and SARS-CoV-2 Co-Infected: 8 TB Only: 4 | TB and SARS-CoV-2 Co-Infected: 3 (38%) TB Only: 1 (25%) | 1.80 (0.12 to 26.2) |
| | Only SARS-CoV-2 diagnosed >28 days after TB | All Unfavorable TB Outcomes | TB and SARS-CoV-2 Co-Infected: 12 TB Only: 4 | TB and SARS-CoV-2 Co-Infected: 3 (25%) TB Only: 1 (25%) | 1.00 (0.07 to 13.64) |
| Gubkina† (2020) [16] | All Participants* (All SARS-CoV-2 diagnosed >28 days after TB) | All Unfavorable TB Outcomes | TB and SARS-CoV-2 Co-Infected: 24 TB Only: 1 | TB and SARS-CoV-2 Co-Infected: 0 (0%) TB Only: 0 (0%) | N/A |
| Gomes (2021) [17] | All Participants* | All Unfavorable TB Outcomes | TB and SARS-CoV-2 Co-Infected: 18 TB Only: 35 | TB and SARS-CoV-2 Co-Infected: 2 (11%) TB Only: 9 (26%) | 0.36 (0.07 to 1.89) |
| | Only SARS-CoV-2 diagnosed within 28 days of TB | All Unfavorable TB Outcomes | TB and SARS-CoV-2 Co-Infected: 2 TB Only: 35 | TB and SARS-CoV-2 Co-Infected: 0 (0%) TB Only: 9 (26%) | N/A |
| | Only SARS-CoV-2 diagnosed >28 days before TB | All Unfavorable TB Outcomes | TB and SARS-CoV-2 Co-Infected: 16 TB Only: 35 | TB and SARS-CoV-2 Co-Infected: 2 (13%) TB Only: 9 (26%) | 0.41 (0.08 to 2.18) |

Abbreviations: TB, tuberculosis; HIV, human immunodeficiency virus; SARS-CoV-2, severe acute respiratory syndrome coronavirus 2; N/A, not applicable.

*Includes participants currently infected and seropositive in the co-infected group.

†In this study, testing for current and previous infection occurred at least 2 months into TB treatment. We did not try to infer timing of infection for the seropositive group.

Further details on treatment failure, death and loss to follow up are presented in Table E, F, and G in S1 Text.

other immunomodulating treatments (n = 290 corticosteroids alone; n = 1 corticosteroids and intravenous immunoglobulin), while 1,281 (81%) did not receive such treatment. Two studies (50%) recruited participants in 2021 and 2022 [21,22]. All studies involved adults only. Most participants were male and approximately half were living with HIV (**Table 1**).

**Primary outcomes.** The first primary outcome of interest, any unfavorable TB treatment outcome, was only assessed in one study [20] with 118 participants, of whom 23 (19%)

received corticosteroids or immunomodulating treatment and 95 (81%) did not (**Table 3**). The crude odds ratio for unfavorable TB treatment outcomes was 10.93 (95% CI 3.03 to 39.39), however this estimate is likely confounded by indication as patients receiving cortico-steroids or immunomodulating treatment were much more likely to be administered supplemental oxygen and/or receive mechanical ventilation, which are proxies for more severe illness and recognized indications for such treatments (**S2 Text**). In subgroup analyses of the timing of SARS-CoV-2 diagnosis in relation to TB diagnosis, estimated odds ratios for poor outcomes were lower for those diagnosed with SARS-CoV-2 within 28 days of TB treatment initiation and higher for those diagnosed > 28 days TB post-initiation (**Table 3**).

The second primary outcome of interest, a composite outcome of severe/critical COVID-19 (requiring ICU admission and/or mechanical ventilation) and/or death, was assessed in one study (**Table 3**) [21]. This study included 806 patients, of whom 117 (15%) received corticoste-roids (no patient received any other immunomodulator) and 689 (85%) did not. The outcome of interest occurred in 60 individuals receiving corticosteroids (51%) and 541 individuals not receiving corticosteroids (79%). After adjustment for demographic and clinical factors, includ-ing use of supplemental oxygen—an indication for steroid use—there was no difference in the risk of severe/critical COVID-19 or death by steroid use (adjusted odds ratio: 1.00, 95% CI 0.59 to 1.70). In subgroup analyses based on HIV (**Table 3**), age, sex, and previous TB history, and circulating variant (**S2 Text**; **S3 Text**), no significant association was found.

**Secondary outcomes.** Few studies reported the secondary outcomes targeted by this review. All-cause mortality was reported in one study, [20] while in-hospital mortality was reported in three studies (**Table 3**) [21–23]. Among studies where statistical adjustment was possible [21,22], findings were inconsistent (**S2 Text**; **S3 Text**). In *Jassat et al*, where use of supplemental oxygen was known, there was no evidence of an effect of corticosteroid use on in-hospital mortality risk (adjusted odds ratio 1.11, 95% CI 0.66 to 1.87) among 806 partici-pants [21]. However, in *Davies et al*, where use of supplemental oxygen was not known, corti-costeroids appeared protective (adjusted hazard ratio 0.49, 95% CI 0.30 to 0.79) among 633 participants—in this study, only those surviving at least 48 hours after admission were included as this was the time it would take for corticosteroids to be prescribed and initiated [22]. Though vaccination status was known, only a small minority of participants were fully vaccinated (5%), preventing any related analysis by vaccination status. Further results are in **Table G in S1 Text** and **S3** Text.

## Overall evidence profile

All included studies were retrospective and observational, with differing designs and patient populations. The certainty of evidence was deemed very low for all outcomes for both the *Coinfection* and *Clinical Management Review* due to very serious risk of bias, serious or very serious inconsistency for different outcomes across studies. Imprecision of effect estimates was serious or very serious, due to the very small patient populations included (**Table H in S1 Text**). References for supplemental files are provided in **S4 Text**.

## Updated search

To determine if any manuscripts with relevant data in the primary publication (i.e., data that we did not need to contact authors to collect) were published since our original search, on April 3, 2023, we repeated our search in the WHO library—which indexes both preprints and published articles and contained all originally identified studies—using the same search strat-egy. We identified 1130 new publications, of which 4 were selected for full-text review [24–27].

**Table 3. Outcomes for select populations in each study evaluating TB patients co-infected with SARS-CoV-2 receiving corticosteroids or other immunomodulating treatment vs. not receiving such treatment (*Clinical Management Review*).**

| Author (Year) | Population Subgroup | Outcome | Participants with Known Outcome | Participants with Outcome (%) | Odds Ratio (95% CI) for Outcome if Receiving Steroids/Immunomodulating Treatment |
|---|---|---|---|---|---|
| The TB/COVID-19 Global Study Group (2022) [20] | All Participants | All Unfavorable TB Outcomes | Receiving Steroids or IM Treatment: 23 No Treatment: 95 | Receiving Steroids or IM Treatment: 20 (87%) No Treatment: 36 (38%) | 10.93 (3.03 to 39.39) |
| | | All-Cause Mortality | Receiving Steroids or IM Treatment: 23 No Treatment: 95 | Receiving Steroids or IM Treatment: 18 (78%) No Treatment: 32 (34%) | 7.09 (2.41 to 20.83) |
| | | Mechanical Ventilation | Receiving Steroids or IM Treatment: 23 No Treatment: 82 | Receiving Steroids or IM Treatment: 11 (48%) No Treatment: 16 (20%) | 3.78 (1.41 to 10.11) |
| | Only SARS-CoV-2 diagnosed within 28 days of TB | All Unfavorable TB Outcomes | Receiving Steroids or IM Treatment: 13 No Treatment: 39 | Receiving Steroids or IM Treatment: 11 (85%) No Treatment: 19 (49%) | 5.79 (1.13 to 29.62) |
| | | All-Cause Mortality | Receiving Steroids or IM Treatment: 13 No Treatment: 39 | Receiving Steroids or IM Treatment: 10 (77%) No Treatment: 19 (49%) | 3.51 (0.84 to 14.73) |
| | | Mechanical Ventilation | Receiving Steroids or IM Treatment: 13 No Treatment: 32 | Receiving Steroids or IM Treatment: 4 (31%) No Treatment: 10 (31%) | 0.98 (0.24 to 3.95) |
| | Only SARS-CoV-2 diagnosed >28 days after TB | All Unfavorable TB Outcomes | Receiving Steroids or IM Treatment: 8 No Treatment: 53 | Receiving Steroids or IM Treatment: 7 (88%) No Treatment: 15 (28%) | 17.73 (2.01 to 156.7) |
| | | All-Cause Mortality | Receiving Steroids or IM Treatment: 8 No Treatment: 53 | Receiving Steroids or IM Treatment: 6 (75%) No Treatment: 11 (21%) | 11.45 (2.03 to 64.77) |
| | | Mechanical Ventilation | Receiving Steroids or IM Treatment: 8 No Treatment: 48 | Receiving Steroids or IM Treatment: 6 (75%) No Treatment: 5 (10%) | 6.45 (1.58 to 26.32) |
| | Only SARS-CoV-2 diagnosed >28 days before TB | All Unfavorable TB Outcomes | Receiving Steroids or IM Treatment: 2 No Treatment: 3 | Receiving Steroids or IM Treatment: 2 (100%) No Treatment: 2 (67%) | N/A |
| | | All-Cause Mortality | Receiving Steroids or IM Treatment: 2 No Treatment: 3 | Receiving Steroids or IM Treatment: 2 (100%) No Treatment: 2 (67%) | N/A |
| | | Mechanical Ventilation | Receiving Steroids or IM Treatment: 2 No Treatment: 2 | Receiving Steroids or IM Treatment: 1 (50%) No Treatment: 1 (50%) | 1.00 (0.02 to 50.40) |

(*Continued*)

**Table 3.** (*Continued*)

| Author (Year) | Population Subgroup | Outcome | Participants with Known Outcome | Participants with Outcome (%) | Odds Ratio (95% CI) for Outcome if Receiving Steroids/Immunomodulating Treatment |
|---|---|---|---|---|---|
| du Bruyn (2021) [23] | All Participants | In-Hospital Mortality | Receiving Steroids or IM Treatment: 8 No Treatment: 7 | Receiving Steroids or IM Treatment: 5 (63%) No Treatment: 1 (14%) | 10.00 (0.78 to 128.77) |
| | Only SARS-CoV-2 diagnosed within 28 days of TB | In-Hospital Mortality | Receiving Steroids or IM Treatment: 7 No Treatment: 6 | Receiving Steroids or IM Treatment: 5 (71%) No Treatment: 1 (17%) | 12.50 (0.84 to 186.3) |
| | Only SARS-CoV-2 diagnosed >28 days after TB | In-Hospital Mortality | Receiving Steroids or IM Treatment: 1 No Treatment: 1 | Receiving Steroids or IM Treatment: 0 (0%) No Treatment: 0 (0%) | 10.00 (0.78 to 128.77) |
| Jassat* (2021) [21] | All Participants | In-Hospital Mortality | Receiving Steroids or IM Treatment: 117 No Treatment: 689 | Receiving Steroids or IM Treatment: 56 (48%) No Treatment: 502 (73%) | 1.11 (0.66 to 1.87) |
| | | Mechanical Ventilation | Receiving Steroids or IM Treatment: 117 No Treatment: 689 | Receiving Steroids or IM Treatment: 2 (2%) No Treatment: 13 (2%) | 0.61 (0.08 to 2.68) |
| | | ICU Admission | Receiving Steroids or IM Treatment: 117 No Treatment: 689 | Receiving Steroids or IM Treatment: 4 (3%) No Treatment: 49 (7%) | 0.70 (0.20 to 2.00) |
| | | ICU Admission, Mechanical Ventilation, or In-Hospital Mortality | Receiving Steroids or IM Treatment: 117 No Treatment: 689 | Receiving Steroids or IM Treatment: 60 (51%) No Treatment: 541 (79%) | 1.00 (0.59 to 1.70) |
| | People living with HIV | In-Hospital Mortality | Receiving Steroids or IM Treatment: 77 No Treatment: 358 | Receiving Steroids or IM Treatment: 37 (48%) No Treatment: 279 (78%) | 1.03 (0.53 to 2.01) |
| | | Mechanical Ventilation | Receiving Steroids or IM Treatment: 77 No Treatment: 358 | Receiving Steroids or IM Treatment: 0 (0%) No Treatment: 2 (1%) | N/A |
| | | ICU Admission | Receiving Steroids or IM Treatment: 77 No Treatment: 358 | Receiving Steroids or IM Treatment: 0 (0%) No Treatment: 13 (4%) | N/A |
| | | ICU Admission, Mechanical Ventilation, or In-Hospital Mortality | Receiving Steroids or IM Treatment: 77 No Treatment: 358 | Receiving Steroids or IM Treatment: 37 (48%) No Treatment: 291 (81%) | 0.85 (0.43 to 1.68) |
| | People without HIV | In-Hospital Mortality | Receiving Steroids or IM Treatment: 22 No Treatment: 281 | Receiving Steroids or IM Treatment: 9 (41%) No Treatment: 192 (68%) | 1.49 (0.48 to 4.54) |
| | | Mechanical Ventilation | Receiving Steroids or IM Treatment: 22 No Treatment: 281 | Receiving Steroids or IM Treatment: 2 (9%) No Treatment: 11 (4%) | 0.83 (0.11 to 3.97) |
| | | ICU Admission | Receiving Steroids or IM Treatment: 22 No Treatment: 281 | Receiving Steroids or IM Treatment: 3 (14%) No Treatment: 36 (13%) | 0.76 (0.16 to 2.64) |
| | | ICU Admission, Mechanical Ventilation, or In-Hospital Mortality | Receiving Steroids or IM Treatment: 22 No Treatment: 281 | Receiving Steroids or IM Treatment: 12 (55%) No Treatment: 219 (78%) | 1.06 (0.32 to 3.30) |

(*Continued*)

**Table 3.** (Continued)

| Author (Year) | Population Subgroup | Outcome | Participants with Known Outcome | Participants with Outcome (%) | Odds Ratio (95% CI) for Outcome if Receiving Steroids/Immunomodulating Treatment |
|---|---|---|---|---|---|
| Davies[†] (2022) [22] | All Participants | In-Hospital Mortality | Receiving Steroids or IM Treatment: 143 No Treatment: 490 | Receiving Steroids or IM Treatment: 18 (13%) No Treatment: 108 (22%) | Hazard Ratio: 0.49 (0.30 to 0.79) |
| | People living with HIV | | Receiving Steroids or IM Treatment: 96 No Treatment: 294 | Receiving Steroids or IM Treatment: 9 (9%) No Treatment: 62 (21%) | Hazard Ratio: 0.39 (0.20 to 0.77) |
| | People without HIV | | Receiving Steroids or IM Treatment: 47 No Treatment: 196 | Receiving Steroids or IM Treatment: 9 (19%) No Treatment: 46 (24%) | Hazard Ratio: 0.59 (0.31 to 1.10) |

Abbreviations: TB, tuberculosis; HIV, human immunodeficiency virus; SARS-CoV-2, severe acute respiratory syndrome coronavirus 2; IM, immunomodulating; ICU, intensive care unit; N/A, not applicable.

*Odds ratios for this study are adjusted based on age, sex, previous tuberculosis history, HIV/ART, and use of supplemental oxygen.

†This study estimates hazard ratios, which are adjusted based on age, sex, HIV, variant wave, full vaccination status, geographic location in Western Cape (Cape Town vs. other), diabetes, hypertension, chronic kidney disease, prior chronic obstructive pulmonary disease, and previous tuberculosis history.

None of these studies reported data in their primary publication that would have been suitable for this review.

## Discussion

Evidence was limited and very low-quality for the impact of SARS-CoV-2 coinfection on TB treatment outcomes. Across studies, the available study data provided no evidence of an association between SARS-CoV-2 infection and end of TB treatment outcomes. This was consistent for secondary outcomes and among identified subgroups.

With respect to the role of corticosteroids or immunomodulating treatment in persons with TB and SARS-CoV-2 infection, evidence was also limited and of very low quality. Data were only available for corticosteroid use, as only one participant in the included studies received a different immunomodulator that was given in conjunction with a corticosteroid. Uncontrolled and unadjusted studies were likely strongly confounded by indication. Where adjustment was possible (i.e., for demographic and clinical factors that influence prescription), there was no evidence of benefit or harm of steroid treatment, with respect to the composite outcome of progression to severe/critical COVID-19 or death [21]. Moreover, risk of misclassification bias due to unrecorded use of steroid treatment was considered high by the study authors.

We found no evidence describing the role SARS-CoV-2 vaccination or current therapeutics (e.g., ritonavir-boosted nirmatrelvir) may play in impacting TB or SARS-CoV-2 outcomes [28]. This is primarily due to the timeframe when studies were conducted (largely pre-vaccination and before strong randomized clinical trial evidence for specific therapeutics emerged), small numbers of participants fully vaccinated where these data were known, and the limited availability of such interventions in the settings studied. We found very limited evidence describing heterogeneity in outcomes between different SARS-CoV-2 variants.

Purpose-built, well-designed studies of the effects of SARS-CoV-2 infection on TB treatment outcomes—and how they may be modulated by prior immunity—are needed.

Prospective designs must use systematic, dual screening for TB and SARS-CoV-2 to reduce risk of misclassification bias while collecting data on prior SARS-CoV-2 infection and vaccination and employing extended follow-up for not only end of treatment outcomes (as is being done in one prospective study) [29], but also consequences of post-TB [30] and post-COVID-19 sequelae [31]. Retrospective designs could use real-world data sources such as linkages between COVID-19 and TB registries, as well as hospitalization, physician claims, prescription drug dispensations, and vital statistics databases, in settings where systematic screening programs were in place for SARS-CoV-2, to evaluate risk of unfavorable TB treatment outcomes by SARS-CoV-2 infection status. To address steroid or other immunomodulating treatment use among persons with SARS-CoV-2/TB coinfection, as well as other SARS-CoV-2 treatments, observational studies designed for causal inference—such as those emulating a target trial [32]—or randomized controlled trials could be employed. To reduce risk of confounding by indication, studies should carefully define inclusion criteria and ensure robust information is collected/available on the timing and occurrence of key events influencing both outcome and exposure (ideally informed by directed acyclic graphs) to permit use of appropriate statistical techniques to address time-varying confounders. These studies should address outcomes beyond effectiveness and include safety, especially considering the multiple medications individuals with TB must take for their treatment and the associated risks of toxicity [33]. For both review questions, future studies should be done in epidemiologically relevant populations, which include those with high-levels of infection- or vaccine-acquired immunity exposed to currently circulating variants.

A key strength of our review is that we were able to obtain additional data from the authors of several studies. However, these data came with substantial limitations. Data shared by authors were generally limited in nature with respect to sample size, patient characteristics, and outcomes assessed, which impacted our ability to address key subgroups (such as pregnant individuals and children) and outcomes identified in the PECO question, as well as statistical power to detect differences. Limited sample sizes among studies with populations suitable to assess the effect of SARS-CoV-2 on TB, and serious risk of confounding by indication among studies with populations suitable to assess steroid use among coinfected individuals were primary limitations for each review question, respectively. An additional limitation is that corticosteroids were the only immunomodulating treatment used in the included studies, and it remains uncertain what the impacts of other treatments may be. Studies generally took place early in the SARS-CoV-2 pandemic and their applicability to present clinical decisions is unclear. Studies included in this review were too heterogeneous to combine in a meta-analysis, limiting our summary of evidence to a narrative synthesis. The evidence was of very low quality, with significant biases, inconsistency, indirectness, and imprecision. In addition, patients included were mostly from inpatient settings, which suggest the findings from this study are not generalizable to most TB patients, the majority of whom are treated in outpatient settings.

In summary, it was unclear whether persons treated for TB with SARS-CoV-2 are at increased risk of unfavorable TB treatment outcomes when compared to those without SARS-CoV-2. It was also unclear whether the use of corticosteroids or other immunomodulating treatments were associated with improved outcomes in persons with TB and SARS-CoV-2 coinfection. Considering the overall low quality of the available evidence, future research is urgently needed to improve our understanding of the risk of unfavorable TB treatment outcomes by SARS-CoV-2 infection status and the optimal management of persons treated for TB with SARS-CoV-2 infection.

## Supporting information

**S1 Text. Supplemental tables and information. Table A in S1 Text.** Search Strategies for Different Databases. **Table B in S1 Text.** PECO Question: In people with tuberculosis disease, does SARS-CoV-2 infection influence outcomes? **Table C in S1 Text.** Adapted Newcastle-Ottawa Scale for Quality Assessment for Cohort and Case-Control Studies. **Table D in S1 Text.** List of Excluded Studies (Non-Duplicate) at Full-Text Stage and Reasons for Not Sharing Data. **Table E in S1 Text.** Mortality Among Select Populations in Each Study Evaluating TB Patients Co-infected with SARS-CoV-2 vs. TB Patients without SARS-CoV-2. **Table F in S1 Text.** Loss to Follow-up Among Select Populations in Each Study Evaluating TB Patients Co-infected with SARS-CoV-2 vs. TB Patients without SARS-CoV-2. **Table G in S1 Text.** Treatment Failure Among Select Populations in Each Study Evaluating TB Patients Co-infected with SARS-CoV-2 vs. TB Patients without SARS-CoV-2. **Table H in S1 Text.** Evidence Profile.
(DOCX)

**S2 Text. Additional details for each study as supplied by authors.**
(DOCX)

**S3 Text. Additional results.**
(DOCX)

**S4 Text. References in supplemental files.**
(DOCX)

## Acknowledgments

The authors are grateful to Genevieve Gore, medical librarian at McGill University, for their support in designing the search strategies. We are grateful for critical reviews of earlier versions of this manuscript by Dick Menzies (McGill University) and Dennis Falzon (World Health Organization). We are also grateful to Lütfiye Kılıç, Erika Mohr-Holland, Petros Isaakidis, Jennifer Furin, Helen Cox, Mario Raviglione, Simone Villa, Catherine Riou, Natalia Yukhimenko, Zulmansyah Zulmansyah, Karen Gomes, Richard Welch, P.S. Rakesh, and contributors to the TB/COVID-19 Global Study Group (Nicolas Casco, Alberto Levi Jorge, Domingo Juan Palmero, Jan-Willem Alffenaar, Justin Denholm, Greg J. Fox, Wafaa Ezz, Jin-Gun Cho, Alena Skrahina, Varvara Solodovnikova, Pierre Bachez, Alberto Piubello, Marcos Abdo Arbex, Tatiana Alves, Marcelo Fouad Rabahi, Giovana Rodrigues Pereira, Roberta Sales, Denise Rossato Silva, Muntasir M. Saffie, Ruth Caamaño Miranda, Viviana Cancino, Monica Carbonell, Catalina Cisterna, Clorinda Concha, Arturo Cruz, Nadia Escobar Salinas, Macarena Espinoza Revillot, Joaquín Farías Valdés, Israel Fernandez, Ximena Flores, Patricia Gallegos Tapia, Ana Garavagno, Carolina Guajardo Vera, Martina Hartwig Bahamondes, Luis Moyano Merino, Eduardo Muñoz, Camila Muñoz, Indira Navarro, Jorge Navarro Subiabre, Carlos Ortega, Sofia Palma, Ana María Pradenas, Gloria Pereira, Patricia Perez Castillo, Mónica Pinto, Rolando Pizarro, Francisco Rivas Bidegain, Patricia Rodriguez, Cristina Sánchez, Angeles Serrano Salinas, Aline Soto, Carolina Taiba, Margarita Venegas, Maria Soledad Vergara Riquelme, Evelyn Vilca, Claudia Villalón, Edith Yucra, Yang Li, Andres Cruz, Beatriz Guelvez, Regina Victoria Plaza, Kelly Yoana Tello Hoyos, Claire Andréjak, François-Xavier Blanc, Samir Dourmane, Antoine Froissart, Armine Izadifar, Frédéric Rivière, Frédéric Schlemmer, Katerina Manika, Nitesh Gupta, Pranav Ish, Gyanshankar Mishra, Samridhi Sharma, Rupak Singla, Zarir F. Udwadia, Boubacar Djelo Diallo, Souleymane Hassane-Harouna, Norma Artiles, Licenciada Andrea Mejia, Francesca Alladio, Fabio Angeli, Andrea Calcagno, Luigi

Ruffo Codecasa, Angelo De Lauretis, Susanna Esposito, Beatrice Formenti, Alberto Gaviraghi, Vania Giacomet, Delia Goletti, Gina Gualano, Alberto Matteelli, Ilaria Motta, Fabrizio Palmieri, Emanuele Pontali, Tullio Prestileo, Niccolò Riccardi, Laura Saderi, Matteo Saporiti, Giovanni Sotgiu, Claudia Stochino, Marina Tadolini, Alessandro Torre, Simone Villa, Dina Visca, Edvardas Danila, Saulius Diktanas, Onno W. Akkerman, Ruy López Ridaura, Fátima Leticia Luna López, Marcela Muñoz Torrico Adrian Rendon, Mahamadou Bassirou Souleymane, Seif Al-Abri, Fatma Alyaquobi, Khalsa Althohli, Edwin Aizpurua, Rolando Gonzales, Julio Jurado, Alejandra Loban, Sarita Aguirre, Rosarito Coronel Teixeira, Viviana De Egea, Sandra Irala, Angélica Medina, Guillermo Sequera, Natalia Sosa, Fátima Vázquez, Félix K. Llanos-Tejada, Selene Manga, Renzo Villanueva -Villegas, David Araujo, Raquel Duarte, Tânia Sales Marques, Victor Ionel Grecu, Adriana Socaci, Olga Barkanova, Maria Bogorodskaya, Sergey Borisov, Andrei Mariandyshev, Anna Kaluzhenina, Tatjana Adzic Vukicevic, Maja Stosic, Darius Beh, Deborah Ng, Catherine W.M. Ong, Ivan Solovic, Keertan Dheda, Phindile Gina, José A Caminero, José Cardoso-Landivar, Maria Luiza De Souza Galvão, Angel Dominguez-Castellano, José-María García-García, Israel Molina Pinargote, Sarai Quirós Fernandez, Adrián Sánchez-Montalvá, Eva Tabernero Huguet, Miguel Zabaleta Murguiondo, Pierre-Alexandre Bart, Jesica Mazza-Stalder, Freya Bakko, James Barnacle, Annabel Brown, Shruthi Chandran, Kieran Killington, Kathy Man, Padmasayee Papineni, Simon Tiberi, Natasa Utjesanovic, Dominik Zenner, Jasie L. Hearn, Scott Heysell, Laura Young) for providing additional data from their studies supporting this evidence synthesis.

## Author Contributions

**Conceptualization:** James C. Johnston, Faiz Ahmad Khan, Kevin Schwartzman, Jonathon R. Campbell.

**Data curation:** Nirma Khatri Vadlamudi, C. Andrew Basham, Giovanni Battista Migliori, Rosella Centis, Lia D'Ambrosio, Waasila Jassat, Mary-Ann Davies, Jonathon R. Campbell.

**Formal analysis:** Nirma Khatri Vadlamudi, C. Andrew Basham, Jonathon R. Campbell.

**Funding acquisition:** Jonathon R. Campbell.

**Investigation:** Nirma Khatri Vadlamudi, C. Andrew Basham, Jonathon R. Campbell.

**Methodology:** Nirma Khatri Vadlamudi, James C. Johnston, Faiz Ahmad Khan, Kevin Schwartzman, Jonathon R. Campbell.

**Project administration:** Jonathon R. Campbell.

**Resources:** Jonathon R. Campbell.

**Supervision:** Jonathon R. Campbell.

**Validation:** Nirma Khatri Vadlamudi, Jonathon R. Campbell.

**Visualization:** Jonathon R. Campbell.

**Writing – original draft:** Nirma Khatri Vadlamudi, Jonathon R. Campbell.

**Writing – review & editing:** C. Andrew Basham, James C. Johnston, Faiz Ahmad Khan, Giovanni Battista Migliori, Rosella Centis, Lia D'Ambrosio, Waasila Jassat, Mary-Ann Davies, Kevin Schwartzman, Jonathon R. Campbell.

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
