## [Decision Letter · Decision Letter 0]

20 Jun 2023

The association of SARS-CoV-2 infection and tuberculosis disease with unfavorable treatment outcomes: a systematic review

PGPH-D-23-00670

Dear Dr. Campbell,

We are pleased to inform you that your manuscript 'The association of SARS-CoV-2 infection and tuberculosis disease with unfavorable treatment outcomes: a systematic review' has been provisionally accepted for publication in PLOS Global Public Health.

Best regards,

Julio Croda, Ph.D, M.D.

Academic Editor

Reviewer Comments (if any, and for reference):

Reviewer's Responses to Questions

**Comments to the Author**

1. Does this manuscript meet PLOS Global Public Health’s publication criteria? Is the manuscript technically sound, and do the data support the conclusions? The manuscript must describe methodologically and ethically rigorous research with conclusions that are appropriately drawn based on the data presented.

Reviewer #1: Yes

Reviewer #2: Yes

2. Has the statistical analysis been performed appropriately and rigorously?

Reviewer #1: Yes

Reviewer #2: Yes

3. Have the authors made all data underlying the findings in their manuscript fully available (please refer to the Data Availability Statement at the start of the manuscript PDF file)?

Reviewer #1: Yes

Reviewer #2: Yes

4. Is the manuscript presented in an intelligible fashion and written in standard English?

Reviewer #1: Yes

Reviewer #2: Yes

5. Review Comments to the Author

Reviewer #1: The research question is well defined and relevant in the current epidemiological context.

Considering that the manuscript reports a systematic review in our analysis we have used the tool AMSTAR 2 to proceed the evaluation (BMJ 2017; 358: j4008).

All AMSTAR 2 domains were adequately followed in this systematic review. Please find attached the complete evaluation based on AMSTAR 2.

Reviewer #2: The authors address a current theme that reflects the concern of most researchers working with tuberculosis.

The methodological rigor followed, as well as the transparency in the presentation of all data, make me comfortable with the results presented, as well as with the conclusions of the study.

6. PLOS authors have the option to publish the peer review history of their article (what does this mean?). If published, this will include your full peer review and any attached files.

**Do you want your identity to be public for this peer review?** For information about this choice, including consent withdrawal, please see our Privacy Policy.

Reviewer #1: **Yes: **Mariangela Ribeiro Resende

Reviewer #2: **Yes: **Roberto D Oliveira

<quillbot-extension-portal></quillbot-extension-portal>
